# A new role for erythropoietin in the homeostasis of red blood cells

Clemente F. Arias [iD] [1,2 ✉], Nuno Valente-Leal[3], Federica Bertocchini[1], Sofia Marques [iD] [3], Francisco J. Acosta[4] & Cristina Fernandez-Arias [iD] [3,5 ✉]

The regulation of red blood cell (RBC) homeostasis is widely assumed to rely on the control of cell production by erythropoietin (EPO) and the destruction of cells at a fixed, species-specific age. In this work, we show that such a regulatory mechanism would be a poor homeostatic solution to satisfy the changing needs of the body. Effective homeostatic control would require RBC lifespan to be variable and tightly regulated. We suggest that EPO may control RBC lifespan by determining CD47 expression in newly formed RBCs and SIRP-$\alpha$ expression in sinusoidal macrophages. EPO could also regulate the initiation and intensity of anti-RBC autoimmune responses that curtail RBC lifespan in some circumstances. These mechanisms would continuously modulate the rate of RBC destruction depending on oxygen availability. The control of RBC lifespan by EPO and autoimmunity emerges as a key mechanism in the homeostasis of RBCs.

[1] Centro de Investigaciones Biológicas (CSIC), Madrid, Spain. [2] Grupo Interdisciplinar de Sistemas Complejos (GISC), Madrid, Spain. [3] Instituto de Medicina Molecular, Universidade de Lisboa, Lisboa, Portugal. [4] Departamento de Ecología, Universidad Complutense de Madrid, Madrid, Spain. [5] Departamento de Immunología, Facultad de Medicina, Universidad Complutense de Madrid, Madrid, Spain. ✉email: tifar@ucm.es; crifer25@ucm.es

Red blood cells (RBCs) undergo a continuous turnover in which aged cells are destroyed in the liver and the spleen and are replaced by new cells formed in the bone marrow. The balance between production and destruction must be tightly regulated to ensure oxygen supply to the tissues and to maintain blood volume and viscosity within physiological ranges. It is well established that RBC production is controlled by erythropoietin (EPO)[1,2]. Low oxygen levels increase the concentration of EPO in the blood, which accelerates the proliferation and differentiation of erythroid precursors in the bone marrow, boosting the number of cells[2,3]. An excess of oxygen inhibits EPO production, which delays the recruitment of new RBCs into the blood[4].

The elimination of RBCs from circulation relies on macrophages of the splenic and hepatic sinusoids, which are capable of recognizing and removing RBCs that have reached a critical species-specific age (around 40 days in mice, 60 days in rats, 70 in cats, 120 in humans, 150 in horses, and 160 in bovines[5–7]). The marked regularity in RBC lifespan among mammal species led to the prevalent idea that RBC lifespan is fixed[8,9].

It has long been accepted that the control of cell production by EPO and the destruction of aged RBCs by sinusoidal macrophages suffice to explain the regulation of RBC homeostasis[10–12]. In this work, we suggest that such a regulatory mechanism based on the control of cell production and a fixed lifespan is a poor homeostatic solution to face changes in oxygen demand or availability. The ability to vary cell lifespan is necessary to rapidly adjust the number of cells if the needs of the body tissues change. From a functional viewpoint, if a fixed lifespan severely constrains homeostasis, we should expect lifespan to be variable. The question naturally arises of whether this is mechanistically possible.

RBC lifespan is normally assumed to be determined by oxidative stress. Permanent exposure to oxygen radicals would cause the progressive deterioration of the RBC membrane[13–16], which would eventually mark the cell as a target for phagocytosis[17]. The duration of RBCs in the blood would depend on the levels of antioxidants they inherit from their erythroid precursors[18–20]. In consequence, the regularity of RBC lifespan among mammals would result from a characteristic, species-specific balance between oxidative stress and antioxidants. Under this model, the duration of RBCs would be already fixed when they first enter the blood, so the homeostatic control of the lifespan of circulating RBCs would be unlikely.

In this work, we suggest an alternative explanation for RBC lifespan. It is well-known that oxidative stress causes the translocation of phosphatidylserine (PS) from the inner to the outer layer of the cell membrane[21–23]. Externalized PS acts as an "eat-me" signal that fosters RBC phagocytosis[17,24,25], which seems to confirm that RBC lifespan is actually governed by oxidative stress. However, the phagocytic activity of sinusoidal macrophages is not only dictated by PS. CD47 delivers "don't-eat-me" signals that counterbalance the effect of PS, preventing the destruction of the RBC[26,27]. Moreover, "eat" and "don't-eat" signals must be conveyed from the RBC membrane to the macrophage cytoplasm. This task relies on specific macrophage receptors that bind PS and CD47, such as Axl, Tim4, or Stabilin-2[24], and SIRP-$\alpha$[28–30] respectively. Without these receptors, sinusoidal macrophages would be unresponsive to RBC signals.

Whereas PS externalization may be driven by oxidative stress[31], the levels of CD47 or SIRP-$\alpha$ can be subject to regulation[32], which provides ample opportunities for RBC lifespan modulation. In this work, we combine mathematical modeling and experiments in mice to test this hypothesis. Our results suggest that the levels of CD47 expression in newly formed cells could predetermine their duration in the blood (as hinted in ref. [33]). Short-term changes in the number of SIRP-$\alpha$ receptors in

macrophages might contribute to continuously fine-tuning this predefined lifespan in circulating RBCs. Furthermore, we suggest that EPO might play a key role in controlling these mechanisms of RBC lifespan determination. High EPO levels might upregulate CD47 in newly formed RBCs and SIRP-$\alpha$ in macrophages, thus lengthening the lifespan of circulating cells, while hyperoxia would have the opposite effect. Remarkably, EPO could also control the initiation of autoimmune responses against healthy RBCs, which would further contribute to rapidly reducing lifespan during hyperoxia. Previous studies have shown the effects that artificially changing lifespan has on the dynamics of RBC populations[34]. Our results suggest that the organism could adopt a similar strategy, varying RBC lifespan, and, consequently, the rate of RBC destruction, to adjust the population of circulating cells depending the conditions of oxygen availability.

This approach opens the way to a better understanding of the physiological adaptations triggered by sports training, altitude acclimation, or spatial flights[35,36]. It also suggests that lifespan imbalances might contribute to worsening clinical conditions such as anemia, myelodysplastic syndromes, or malaria infections[37,38].

## Results

**Effect of constant and adaptive lifespan on the homeostasis of red blood cells.** In this section, we will explore how fixed and variable lifespans affect RBC dynamics. To do that, let us first assume a constant lifespan $\bar{L}$. Under this assumption, the cells present in the blood at any time $t$ are those that have been formed between $t - \bar{L}$ and $t$, since all the RBCs formed before $t - \bar{L}$ are no longer alive. If $r(t)$ denotes the number of RBCs at time $t$, we have that:

$$r(t) = \int_{t-\bar{L}}^{t} r_\tau(t) d\tau, \qquad (1)$$

where $r_\tau(t)$ is the number of RBCs born at time $\tau$ that are in the blood at time $t$. Labeling the production of RBCs at time $t$ as $p(t)$ and assuming that RBCs can also be destroyed through non-homeostatic pathways at rate $\mu \geq 0$ (eg. through eryptosis, hemorrhages, infections, hemolysis, ...), we can write:

$$\begin{cases} r'_\tau(t) = -\mu r_\tau(t) \\ r_\tau(\tau) = p(\tau) \end{cases}, \text{ for } \tau \leq t \leq \tau + \bar{L}, \qquad (2)$$

Integrating this equation, we get:

$$r_\tau(t) = p(\tau) e^{-\mu(t-\tau)}, \text{ for } \tau \leq t \leq \tau + \bar{L} \qquad (3)$$

Hence, Equation (1) can be written as:

$$r(t) = \int_{t-\bar{L}}^{t} p(\tau) e^{-\mu(t-\tau)} d\tau, \qquad (4)$$

Therefore, the dynamics of the RBC population are given by:

$$r'(t) = p(t) - e^{-\mu\bar{L}} p(t - \bar{L}) - \mu r(t). \qquad (5)$$

Assuming that RBC lifespan $L(t)$ is variable and using the same arguments, we get:

$$r'(t) = p(t) - e^{-\mu L(t)} p(t - L(t))(1 - L'(t)) - \mu r(t) \qquad (6)$$

Equations (5) and (6) show that constant and variable lifespans differ in their effect on the rate of cell destruction. To clarify this point, let us assume that $\mu = 0$ (i.e., that RBCs are only destroyed through homeostatic pathways). The dynamics of the population can be written as:

$$r'(t) = p(t) - d(t), \qquad (7)$$

where $d(t)$ is the rate of RBC destruction at time $t$.

If lifespan is constant $\left(L(t) = \bar{L}\right)$, then:

$$d(t) = p(t - \bar{L}) \qquad (8)$$

In this scenario, the cells that are phagocytized at a given time $t$ are those that were formed at time $t - \bar{L}$. In consequence, the rate of cell destruction is univocally determined by past rates of cell production. Conversely, changing lifespan at time $t$ (i.e., making $L'(t) \neq 0$) instantly changes the rate of RBC destruction:

$$d(t) = p(t - L(t))(1 - L'(t)). \qquad (9)$$

Therefore, in populations with constant lifespans, the number of cells can only vary through changes in the rate of cell production whereas populations with variable lifespans can also resort to changes in the rate of cell destruction. To understand the consequences of this difference on the dynamics of RBC populations, we will next formulate a model for the rate of cell production. The regulation of RBC production is a complex process in which EPO levels control the proliferation and differentiation of erythroid precursors in the bone marrow. From a modeling perspective, the role of EPO can be implemented either explicitly (see for instance[39,40]), or implicitly, through its resultant effects on the formation of new RBCs (as in ref. [34]). For the sake of simplicity, we will use the latter strategy, and will not explicitly consider the dynamics of EPO or its effect on precursor cells. Despite its mechanistic complexity, the underlying logic of EPO regulation can be described in simple terms: it increases production when the number of cells is below the needs of the body and decreases production otherwise. This logic can be simply modeled by the following equation:

$$p'(t) = \lambda(K(t) - r(t)), \qquad (10)$$

where $K(t)$ is the number of RBCs needed at time $t$ to ensure the oxygenation of the body tissues, and $\lambda$ is a positive parameter. Putting equations (5) and (10) together, the dynamics of RBCs populations with constant lifespans can be modeled as follows:

$$\begin{cases} r'(t) = p(t) - e^{-\mu \bar{L}} p(t - \bar{L}) - \mu r(t) \\ p'(t) = \lambda(K(t) - r(t)). \end{cases} \qquad \text{(Model 1)}$$

Finally, to simulate the dynamics of populations with variable lifespans, we will use the following model:

$$L''(t) = \rho\left(\bar{L} - L(t)\right) - \sigma L'(t) + \omega(K(t) - r(t)), \qquad (11)$$

for positive parameters $\rho$, $\sigma$, and $\omega$.

This model can be reformulated in terms of first-order differential equations as follows:

$$\begin{cases} L'(t) = v(t) \\ v'(t) = \rho\left(\bar{L} - L(t)\right) - \sigma L'(t) + \omega(K(t) - r(t)). \end{cases} \qquad (12)$$

The rationale for the use of this model is that it simulates variable lifespans as oscillating around a reference fixed value $\bar{L}$, which allows comparing its dynamics with that resulting from a constant lifespan $\bar{L}$. Also, in this model, lifespan is adaptive, i.e., it changes depending on the difference between the number of RBCs ($r(t)$) and the needs of the body ($K(t)$). Finally, parameter $\rho$ allows comparing different scenarios of lifespan flexibility, with greater values of this parameter corresponding to more flexible lifespans. Using this model, the dynamics of RBC populations

with variable lifespans can be modeled as

$$\begin{cases} r'(t) = p(t) - e^{-\mu L(t)} p(t - L(t))(1 - L'(t)) - \mu r(t) \\ p'(t) = \lambda(K(t) - r(t)) \\ L'(t) = v(t) \\ v'(t) = \rho\left(\bar{L} - L(t)\right) - \sigma L'(t) + \omega(K(t) - r(t)). \end{cases}$$

$$\text{(Model 2)}$$

The previous models allow comparing the effects of constant and variable lifespans on RBC homeostasis. Numerical simulations of Model 1 show that populations with constant lifespans exhibit delayed responses if RBC demand falls and tend to overcompensate, which creates long-lasting oscillations in the number of cells (Fig. 1a). In these circumstances, diminished RBC production does not suffice to contract the population rapidly enough (Fig. 1b). Increasing RBC demand triggers symmetrical behaviors, with longer lifespans creating greater delays and more pronounced oscillations (Fig. 1c, d).

In view of the previous results, shorter lifespans seem better homeostatic solutions since they reduce the inertia of the population, providing faster adaptations to changes in RBC demand. However, shorter lifespans also entail a greater cost for the organism, since they require higher rates of RBC production to satisfy a given demand $\bar{K}$. From Model 1, the rate of RBC production at the equilibrium ($\bar{p}$) in the absence of non-homeostatic mortality (i.e., $\mu = 0$) is given by:

$$\bar{p} = \frac{\bar{K}}{\bar{L}}, \qquad (13)$$

which is greater for lower values of $\bar{L}$. The non-homeostatic mortality of RBCs entails an additional cost for the organism. In this case ($\mu > 0$), the rate of cell production becomes:

$$\bar{p} = \frac{\bar{K}}{\bar{L}} \left( \frac{\mu \bar{L}}{1 - e^{-\mu \bar{L}}} \right), \qquad (14)$$

which is also larger for shorter lifespans.

In summary, Model 1 evidences that lifespan affects two different dimensions of RBC homeostasis: the cost of cell production and the inertia of the population to changes in RBC demand. Under a constant lifespan, both aspects are inversely related: reducing the cost of production increases inertia and vice versa. Model 2 shows that flexible RBC lifespans uncouple both dimensions, creating less inertia and providing better homeostatic adjustments (Fig. 1e, f) with smaller changes in RBC production (Fig. 1g, h). Constant and variable lifespans also differ greatly in the case of hemorrhages. In populations with constant lifespans, rising cell production does not suffice to compensate for cell loss and the system oscillates sharply before returning to normality (Fig. 1j). Variable lifespans accelerate the recovery of the population (Fig. 1k), with more flexible lifespans entailing lower hemorrhage-derived costs, i.e., smaller increases in RBC production (Fig. 1l).

Model 2 assumes that lifespan is flexible and adaptive, i.e., it changes depending on the difference between RBC demand and availability. Figure 1 shows that this logic confers a remarkable homeostatic plasticity. The question arises of whether adaptive RBC lifespans are within reach of the organism. Published evidence suggests that this might be the case. In particular, it has been observed that EPO may bind circulating RBCs, thereby improving their viability, which would increase their lifespan and reduce the rate of cell destruction in the blood[41]. In the following sections, we will explore other mechanisms that could participate in the regulation of RBC lifespan by EPO.

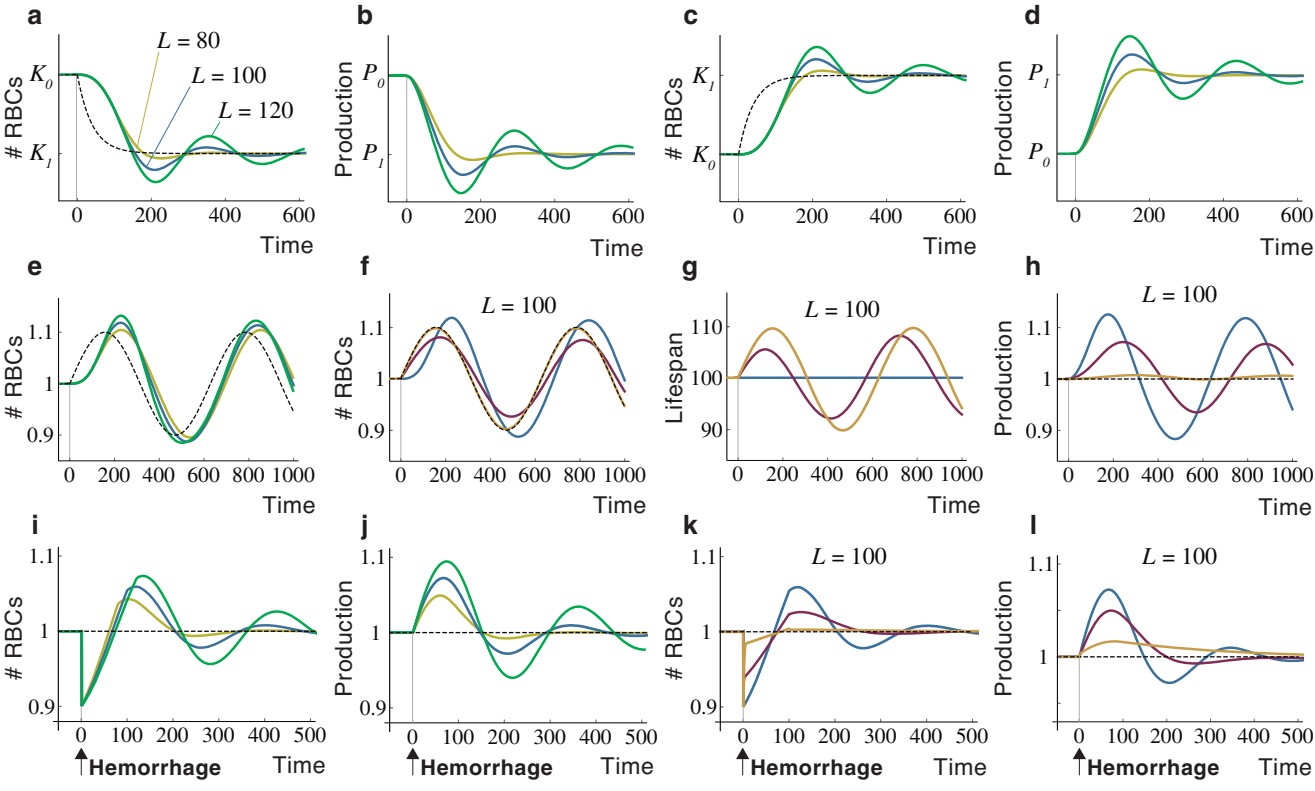

**Fig. 1 Dynamics of RBC populations under constant and variable lifespans.** Response of the number (**a**) and production (**b**) of RBCs with constant lifespans to a drop in demand. RBC production in each scenario is normalized to its initial value. Dynamics of the number (**c**) and production (**d**) of RBCs after a rise in demand under constant lifespans. Response to RBC populations with constant (**e**) and flexible (**f**) lifespans to oscillations in the demand. (The behavior of a population with a constant lifespan $L = 100$ is shown in blue for comparison in **f**). Changes in RBC lifespan (**g**) and production (**h**) induced by the variations in RBC demand shown in **f**. Greater changes in lifespan provide better homeostatic adjustments (as shown in **f**) with smaller variations in RBC production. Effect of hemorrhages on the number (**i**) and production (**j**) of RBCs with constant lifespans. Response of the number (**k**) and production (**l**) of RBCs with flexible lifespan to hemorrhages. The dynamics of RBCs with a constant lifespan are shown in blue for comparison. $L$ = lifespan. Model parameters: $\lambda = 2 \times 10^{-4}$, $\mu = 0$, $\rho = 5$ (yellow line in Fig. 1) and 0.2 (red line), $\sigma = 0.1$, $\omega = 10^{-3}$, and $\bar{K} = 10^5$. Parameters have been chosen arbitrarily to illustrate the behavior of Models 1 and 2. Axes units are also arbitrary.

**Mechanisms of red blood cell lifespan determination.** RBCs passing through the splenic and hepatic sinusoids engage in transient cell-to-cell interactions with resident macrophages[24]. These interactions determine the fate of RBCs in two different ways. Sinusoidal macrophages phagocytize RBCs in which the effect of PS outbalances that of CD47[30]. Splenic macrophages also remove RBCs whose CD47 expression is below a critical level, regardless of the amount of PS in their membrane[42]. In this case, splenic macrophages activate an autoimmune response that induces the production of anti-RBC antibodies[27,33,42–44]. (The consequences of this response will be addressed in later sections.) The previous conditions control two alternative phagocytosis pathways that differ in their consequences on the subsequent macrophage behavior. We will refer to both phagocytosis pathways as silent (meaning that it does not trigger autoimmunity) and immune, respectively[33].

Denoting by $s_E$ and $s_D$ the number of eat and don't-eat signals delivered by PS and CD47 to the macrophage, the conditions that trigger RBC phagocytosis can be can be expressed as

$$s_E - s_D \geq T_S \qquad \text{(Condition 1)}$$

and

$$s_D \leq T_I, \qquad \text{(Condition 2)}$$

for certain values $T_S$ (silent threshold) and $T_I$ (immune threshold). The expression of PS and CD47 changes with the age of the RBC[45]: PS increases with time[46,47] and CD47 disappears

progressively from the membrane[26,48]. These dynamics can be modeled as follows[33]:

$$\begin{cases} D(t) = D_0 e^{-\alpha t} \\ E(t) = E_0 + \beta t \\ D(0) = D_0 \\ E(0) = E_0, \end{cases} \qquad (15)$$

where $t$ is the age of the RBC, $E(t)$ and $D(t)$ denote PS and CD47 expression, respectively, and $\alpha$ and $\beta$ are positive parameters (see ref. [33] for further details about the rationale for the choice of these equations and the meaning of the parameters) (Fig. 2a).

The number of eat and don't-eat signals perceived by a macrophage not only depend on how much PS and CD47 is in the RBC surface but also on how many PS and CD47 receptors it has its membrane ($r_E$ and $r_D$ respectively). This dependence can be modeled as follows:

$$\begin{cases} s_D = \delta_D r_D D(t) \\ s_E = \delta_E r_E E(t), \end{cases} \qquad (16)$$

where $\delta_E$ and $\delta_D$ are positive parameters. For the sake of simplicity and without loss of generality, we will take $\delta_E = \delta_D = 1$.

Equations (15) and (16), together with Conditions 1 and 2, model the outcome of RBC/macrophage interactions. Numerical simulations of this model show that variations in the profile of macrophage receptors or in the dynamics of PS and CD47 in the

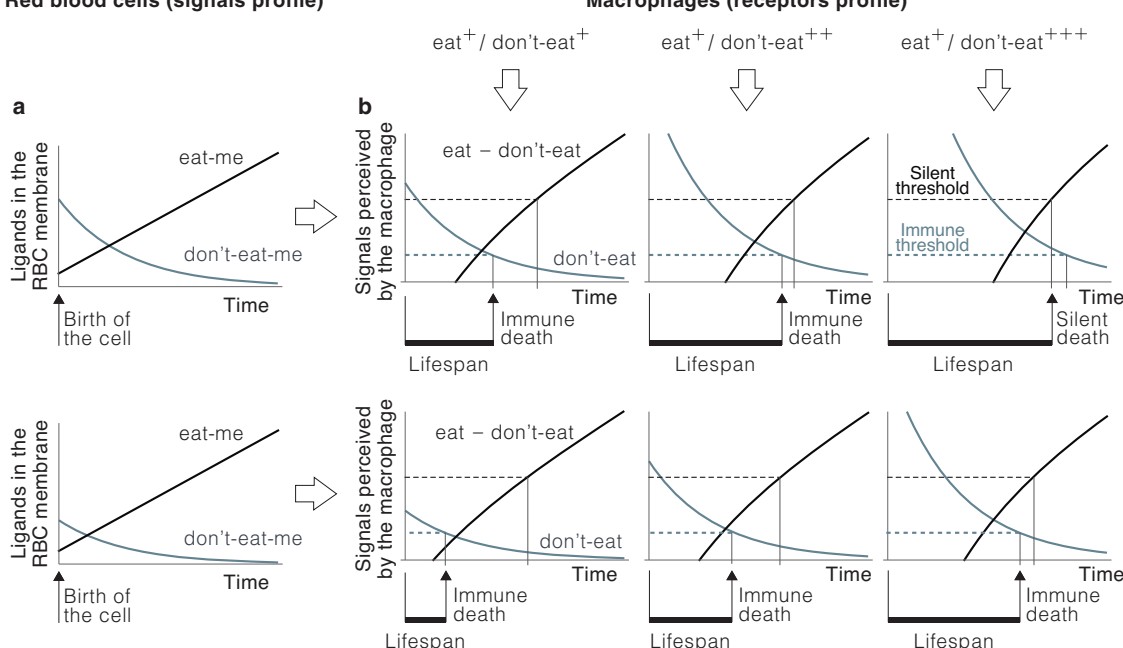

**Fig. 2 Variability of RBC fate and lifespan. a** Dynamics of eat-me and don't-eat-me signals in two RBCs that differ in their initial expression of the latter according to Equation (15). **b** Outcomes of the interaction of RBCs in A with macrophages expressing different levels of don't-eat receptors. RBCs undergo the silent pathway if the difference between the number of "eat" and "don't-eat" signals perceived by the macrophage is above a critical threshold (silent threshold) and the immune pathway if the number of "don't" eat signals does not reach a minimum value (immune threshold). Macrophages with low or medium levels of don't-eat receptors (left and middle column, respectively) phagocytize both RBCs through the immune pathway. However, the timing of the phagocytosis is different, which translates into heterogeneous lifespans. If macrophages express more don't-eat receptors (right column) the lifespan of both RBCs is longer. In this case, the first one (top line) is now phagocytized through the silent pathway. Model parameters: $\alpha = 0.15$, $\beta = 0.25$, $E_0 = 0.6$, $D_0 = 4$ and 2, $r_E = 10$, and $r_D = 10, 20$, and 40, $\delta_E = \delta_D = 1$. Parameters have been chosen arbitrarily to show the qualitative behavior of Equations (15) and (16). To simplify the graphs, axes units are not shown.

RBC entail sharp differences both in the phagocytosis pathway undergone by RBCs and their lifespans (Fig. 2b).

These results show that assuming oxidation-driven PS externalization as the main cause of RBC phagocytosis is overly simplistic (we remark that RBCs in Fig. 2 do not differ in PS expression). They also show that RBC lifespan is unlikely to be fixed. This would require that the number of receptors be identical in all sinusoidal macrophages at all times. Also, CD47 and PS dynamics should be equal in all RBCs. Otherwise, RBC lifespan is necessarily variable. Can this variability be used by the organism a homeostatic mechanism? We address this question in the next section.

**Role of adaptive lifespan in the homeostasis of red blood cells.** Taking equation (16) into account, Conditions 1 and 2 can be rewritten as

$$r_E E(t) \geq T_S + r_D D(t) \qquad \text{(Condition 1)}$$

and

$$r_D D(t) \leq T_I. \qquad \text{(Condition 2)}$$

From the previous expressions, it is possible to determine at what age are RBCs destroyed by macrophages. Denoting by $t_S$ and $t_I$ the times at which RBCs are phagocytized through the silent and the immune pathways, respectively, we have that:

$$t_S = t \text{ such that } r_E E(t) = T_S + r_D D(t)$$

and:

$$t_I = t \text{ such that } r_D D(t) = T_I$$

Replacing $E(t)$ and $D(t)$ by their values given by Equation (15) gives explicit expressions for $t_S$ and $t_I$:

$$\begin{cases} t_S = \frac{T_S - r_E E_0}{\beta r_E} + \frac{1}{\alpha} W\left( \frac{\alpha r_D D_0}{\beta r_E} e^{\frac{\alpha(r_E E_0 - T_S)}{\beta r_E}} \right) \\ t_I = \frac{1}{\alpha} ln\left( \frac{r_D D_0}{T_I} \right) \end{cases} \qquad \text{(Model 3)}$$

where $W(x)$ is the Lambert function.

When a RBC of age $t$ interacts with hepatic or splenic macrophages, it is phagocytized through the silent pathway if $t = t_S < t_I$ and through the immune pathway if $t = t_I < t_S$. If $t < t_I$ and $t < t_S$, the RBC continues to circulate in the blood. According to Model 3, these conditions depend on how much PS and CD47 the RBC expressed when it was formed ($E_0$ and $D_0$) and on the number of PS and CD47 receptors ($r_E$ and $r_D$) in the macrophages it encounters when passing through the sinusoids. To illustrate the explicative power of our approach, we will focus on how $D_0$ (CD47 expression in newly formed RBCs) and $r_D$ (SIRP-$\alpha$ expression in macrophages) determine the fate and lifespan of RBCs. To do that, hereafter, we will assume that $E_0$ and $r_E$ are constant.

In this scenario, for a given level of SIRP-$\alpha$ expression in macrophages ($r_D$), the fate of RBCs only depends on $D_0$. For RBCs with high values of $D_0$, we have that $t_S < t_I$, so they are phagocytized through the silent pathway. RBCs with $D_0$ below a certain threshold (labeled as autoimmunity threshold) verify that $t_I < t_S$, so they undergo the immune pathway instead. Since each RBC can only by phagocytized once, the first condition of be fulfilled determines its fate. Therefore, RBC lifespan is given by $L = \min(t_S, t_I)$ (see Fig. 3a). From this result, it follows that the

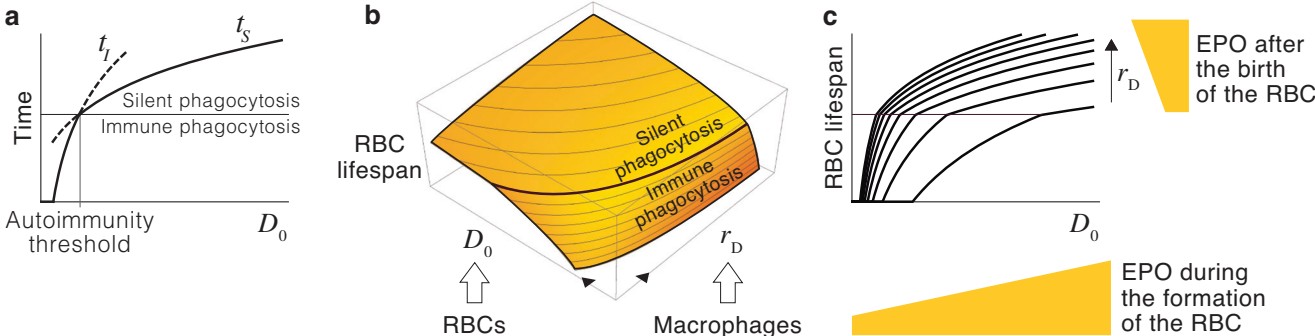

**Fig. 3 Homeostatic control of adaptive RBC lifespan. a** Effect of $D_O$ on the fate and lifespan of RBCs. **b** Combined effect of $D_O$ and $r_D$ on the fate and lifespan of RBCs. The black thick line represents the autoimmunity threshold. **c** We hypothesize that (i) the concentration of EPO during the formation of the cell determines how much CD47 it expresses when it enters the blood ($D_O$), and (ii) the concentration of EPO during the life of the RBC controls SIRP-$\alpha$ expression in macrophages ($r_D$).

lifespan of RBCs is predetermined by their level of CD47 expression when they egress the bone marrow. This predefined lifespan can be modulated once they are in circulation by changes in the number of SIRP-$\alpha$ receptors in macrophages (Fig. 3b).

We suggest that CD47 and SIRP-$\alpha$ define a mechanism of adaptive lifespan that determines the duration of RBCs depending on the difference between RBC demand and availability. As shown in Fig. 1, this is an effective homeostatic strategy to control RBC populations. We hypothesize that this mechanism is regulated by EPO. In particular, we suggest that EPO conditions the expression of CD47 in newly formed RBCs and of SIRP-$\alpha$ in macrophages (Fig. 3c). Under this hypothesis, RBCs are sensitive to EPO in two different ways. The levels of EPO during the development of RBCs in the bone marrow determine their expected lifespan by defining their level of CD47 expression when they enter circulation (as suggested in ref. [33]). Once in the blood, EPO-induced changes in the number of macrophage receptors continuously fine-tune this predefined lifespan.

Under our hypothesis, the rise in the concentration of EPO in response to low oxygen levels would lengthen lifespan, reducing the rate of RBC destruction. This effect, together with the EPO-induced boost in RBC production would increase the number of cells in the blood (see Equation (6)). Conversely, low EPO levels would shorten RBC lifespan, increasing the destruction of RBCs in the liver and the spleen and, therefore, reducing the size of the population. The effect of EPO on CD47 and SIRP-$\alpha$ would also lead to changes in the autoimmunity threshold (Fig. 3c). Low EPO levels would increase the likelihood of immune phagocytosis, which could eventually trigger anti-RBC autoimmune responses. This would accelerate the destruction of RBCs that are no longer needed by the organism. In the next section, we will explore the homeostatic role of this mechanism of EPO-mediated autoimmunity.

**Role of autoimmunity in the homeostasis of red blood cells.** The phagocytosis of RBCs with low CD47 expression triggers autoimmune responses that target circulating RBCs[27,42–44]. Under our hypothesis, this is likely to occur when EPO levels are low (Fig. 4a), a clear indicator of an excess of oxygen in the tissues. We suggest that autoimmunity is a homeostatic strategy to rapidly reduce the number of RBCs in these circumstances[33]. It has been suggested that immunity might play similar roles in the homeostatic control of other tissues[49,50]. To illustrate the operation of this mechanism in the particular case of RBC populations, will model how auto-antibodies affect the fate and lifespan of RBCs.

Opsonized RBCs are recognized by macrophages through specific membrane receptors that bind the $F_c$ region of antibodies. These receptors act as eat signals that foster the phagocytosis of opsonized cells[30]. It has been observed that CD47 inhibits phagocytosis of opsonized RBCs in a dose-dependent manner[51], which implies that macrophages integrate opsonization-derived signals with the rest of eat and don't-eat signals delivered by the membrane of the RBC[30,52]. This integration that can be modeled as follows:

$$\begin{cases} s_D = \delta_D r_D D(t) \\ s_E = \delta_E r_E E(t) + \delta_A, \end{cases} \quad (17)$$

where $\delta_A$ is a positive parameter that represents the degree of opsonization of the RBC.

From Conditions 1 and 2 above, the lifespan of opsonized RBCs is given by:

$$\begin{cases} t_S = \frac{T_S - r_E E_0 - \delta_A}{\beta r_E} + \frac{1}{\alpha} W\left(\frac{\alpha r_D D_0}{\beta r_E} e^{\frac{\alpha(r_E E_0 - T_S + \delta_A)}{\beta r_E}}\right) \\ t_I = \frac{1}{\alpha} ln\left(\frac{r_D D_0}{T_I}\right) \\ L = \min(t_S, t_I), \end{cases} \quad \text{(Model 4)}$$

where $W(x)$ is the Lambert function.

According to Model 4, anti-RBC autoimmune responses does not cause the indiscriminate destruction of opsonized cells. Instead, they shorten the lifespan of circulating cells in proportion to their degree of opsonization (Fig. 4b). We suggest that this provides a homeostatic mechanism, regulated by EPO, intended to provoke a controlled reduction in the number of RBCs.

Under this assumption, diminished EPO levels caused by an excess of oxygen would downregulate SIRP-$\alpha$ receptors in macrophages, shortening RBC lifespan, and inducing the immune phagocytosis of some RBCs. This would trigger anti-RBC autoimmune responses, which would further reduce the lifespan of circulating cells. The intensity of these responses would depend on the number of RBCs phagocytized through the immune pathway at any time, which in turn would be proportional to the drop in the concentration of EPO.

The contraction of the RBC population caused by shortened lifespan would restore oxygen and EPO levels to normal. This would hinder the immune phagocytosis of non-opsonized RBCs (Fig. 4a). Opsonized RBCs would also be less likely to undergo this phagoytosis pathway (Fig. 4c). Eventually, all RBCs would be removed through the silent pathway, ending the production of anti-RBC antibodies. Therefore, homeostatic autoimmunity would contribute to a controlled reduction of the number of RBCs. This would be a reversible process, operating only while EPO levels remain below normal.

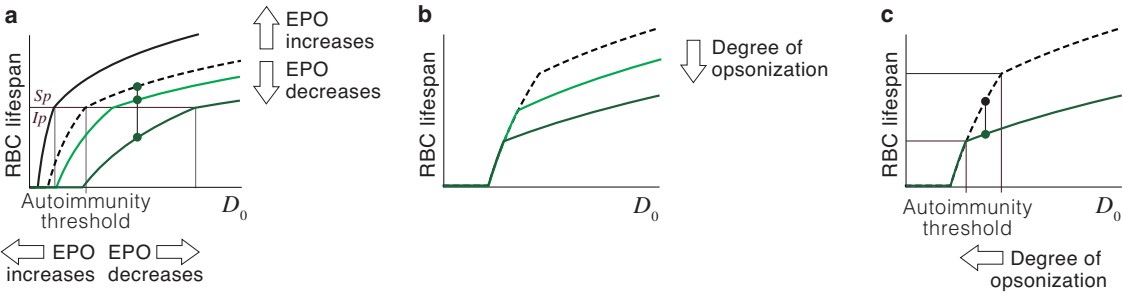

**Fig. 4 Role of autoimmunity in RBC homeostasis. a** An increase in EPO prolongs the duration of RBCs (from the dashed line to the solid black line) and displaces the autoimmune threshold towards lower values of $D_O$, which reduces the likelihood of immune phagocytosis. A drop in the levels of EPO entails shorter lifespans (from the dashed line to the solid light green line). For very low EPO levels, RBCs predestined to undergo silent phagocytosis are removed through the immune pathway instead (dark green line). **b** According to Model 4, auto-antibodies reduce the lifespan of RBCs, an effect that increases with the degree of opsonization. **c** Opsonization also displaces the autoimmunity threshold towards greater values of $D_O$. RBCs that would undergo the immune pathway (black circle) are destroyed through the silent pathway when opsonized (green circle). This implies that the lifespan of opsonized RBCs is shorter but they are less likely to be phagocytized through the immune pathway. The dashed black line in B and C corresponds to non-opsonized RBCs.

**The role of EPO in the adaptive lifespan of red blood cells**. In the previous section, we hypothesized that EPO controls RBC lifespan by modulating the expression of key molecules on RBCs and macrophages. According to Model 2, this would confer the organism effective control over the number of RBCs in the blood. In this section, we provide empirical evidence to support the role of EPO in determining RBC lifespan.

It is well known that low oxygen availability rises the concentration of EPO, which in turn increases the rate of RBC production in the bone marrow[53]. We suggest that this effect is accompanied by an increase in RBC lifespan that reduces the rate of cell destruction (see Equation (6)) and contributes to a net increase in the number of cells (Fig. 1). Conversely, a drop in the concentration of EPO would shorten lifespan and trigger the production of auto-antibodies against circulating RBCs, thus increasing the rate of cell destruction. To test these hypotheses, we treated mice with EPO (see Methods). As expected, EPO raised the rate of RBC production, increasing the percentage of reticulocytes in the blood (Fig. 5a). In agreement with our hypotheses, this effect was accompanied by the upregulation of CD47 in the reticulocytes of EPO-treated mice (Fig. 5b) and also of SIRP-$\alpha$ receptors in the sinusoidal macrophages of the liver (Fig. 5c). According to our model of RBC lifespan determination, the upregulation of SIRP-$\alpha$ would amplify the perception of "don't-eat" signals by macrophages, reducing their phagocytic activity. In agreement with this prediction, RBC phagocytosis was sharply reduced in macrophages treated in vitro with EPO (Fig. 5d). Taken together, these results suggest that increasing EPO levels lengthens RBC lifespan.

Then, we analyzed the effects of hyperoxia (see Methods). As expected under our hypotheses, these conditions did not only lower the percentage of reticulocytes in the blood (Fig. 5e) but also their levels of CD47 expression (Fig. 5f). Importantly, CD47 expression was not changed in the RBCs formed before the hyperoxia treatment (Fig. 5g), which shows that EPO only affects CD47 expression in newly formed RBCs. Also in agreement with our hypotheses, hyperoxia downregulated the number of SIRP-$\alpha$ receptors in macrophages (Fig. 5h) and led to a dramatic increment in the production of anti-PS antibodies (Fig. 5i). These antibodies opsonize RBCs, facilitating their phagocytosis by sinusoidal macrophages[38]. The previous results support the hypothesis that the organism responds to an excess of oxygen in the tissues by shortening the RBC lifespan.

According to Model 3, RBCs formed with low CD47 expression during hyperoxia (Fig. 5f) should exhibit shorter lifespans as compared to control RBCs (Fig. 3). This prediction

was confirmed by monitoring the fate of RBCs in the blood after returning to normal oxygen levels. RBCs formed in hyperoxia underwent a three-fold increase in their rate of removal as compared to control mice (Fig. 5j). In contrast, the rate of removal of RBCs formed before hyperoxia did not show any difference with RBCs from control mice (Fig. 5k), as should be expected from their similar levels of CD47 expression (Fig. 5g).

Overall, these data suggest that EPO could determine the expression of CD47 and SIRP-$\alpha$ receptors in newly formed RBCs and macrophages respectively, thereby regulating the lifespan of circulating RBCs. The production of anti-RBC antibodies in hyperoxia could further contribute to shortening RBC lifespan. Therefore, EPO could not only control the rate at which RBCs are produced in the bone marrow but also the rate at which they are removed by sinusoidal macrophages.

## Discussion
The current paradigm of RBC homeostasis is based on two main assumptions: EPO regulates the production of new cells in the bone marrow[12,54], and sinusoidal macrophages remove RBCs from the blood when they attain a certain species-specific age[55]. Qualitatively, these elements seem to provide a sound explanation for the control of RBC populations. The rate of cell production would be determined by a feedback mechanism that adjusts blood EPO levels depending on the availability of oxygen. Oxidative stress would progressively deteriorate RBCs, and the appearance of senescence signs would allow macrophages to identify and destroy aged cells (Fig. 6a).

In this work, we show that a quantitative formulation of this model reveals critical limitations that are not obvious in its qualitative counterpart, and suggest an alternative view of RBC homeostasis. We identify adaptive lifespan as a key regulatory mechanism to ensure a homeostatic plasticity impossible to achieve with fixed lifespans. Previous studies have shown that EPO might increase RBC lifespan by delaying senescence[41]. We suggest that EPO could also regulate RBC lifespan through its effects on CD47 and SIRP-$\alpha$ expression and through EPO-induced autoimmunity. These elements would shape a simple mechanism allowing to adapt lifespan (and consequently the rate of RBC destruction) depending on the conditions of oxygen availability (Fig. 6b).

The full validation of this model of homeostatic regulation of RBC lifespan will require further empirical studies both in humans and other mammalian species. At a molecular scale, the experimental manipulation of key factors such as CD47 or SIRP-

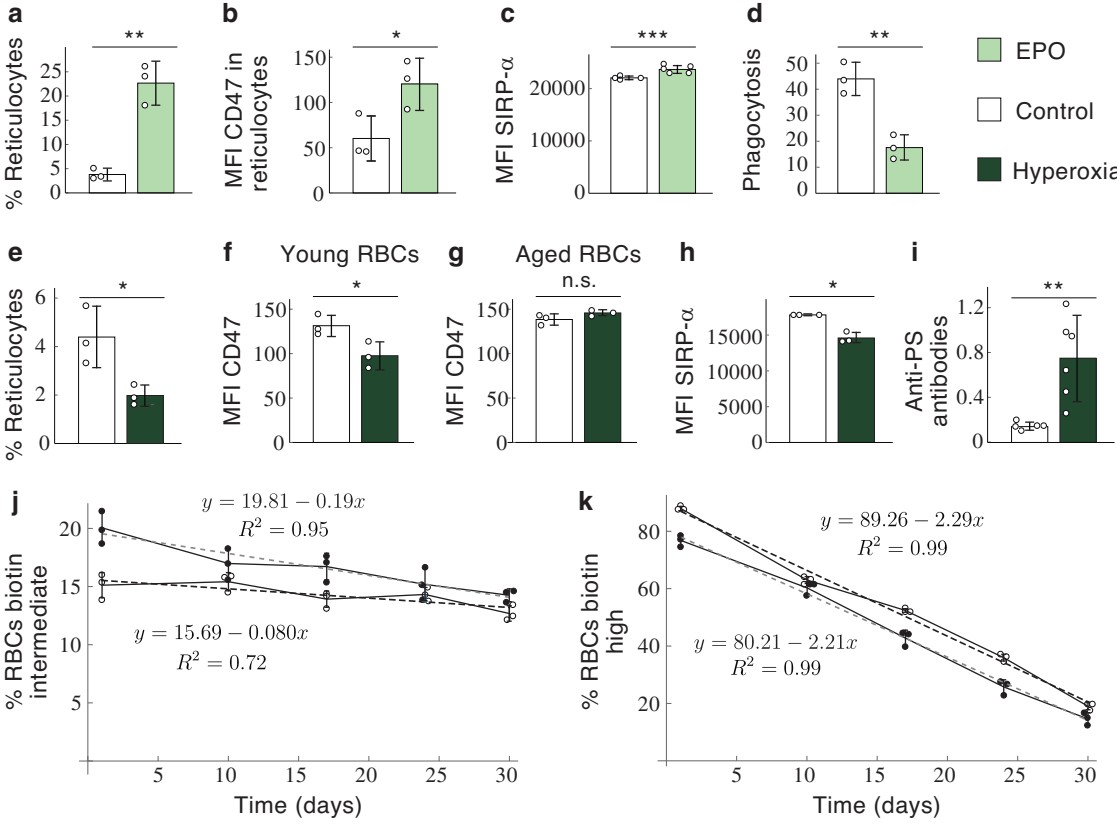

**Fig. 5 Role of EPO in RBC lifespan determination.** In vivo effects of EPO on the production of reticulocytes ($n = 3$) (**a**), CD47 expression in newly formed RBCs ($n = 3$) (**b**), and number of SIRP-$\alpha$ receptors in macrophages of the hepatic sinusoids ($n = 4$ control and $n = 5$ EPO mice) (**c**). **d** In vitro effect of EPO on the phagocytosis of RBCs by sinusoidal macrophages ($n = 3$). **e–i** In vivo effects of hyperoxia on the production of reticulocytes ($n = 3$) (**e**), CD47 expression in RBCs formed during (biotin intermediate) and before (biotin high) hyperoxia ($n = 3$) (**f** and **g**, respectively), SIRP-$\alpha$ receptors in macrophages of the hepatic sinusoids ($n = 3$) (**h**), and production of anti-RBC antibodies ($n = 4$ control mice and $n = 6$ hyperoxia mice) (**i**). **j**, **k** Monitoring of biotin intermediate and biotin high RBCs after the hyperoxia treatment ($n = 3$). Solid dots: hyperoxia mice; white dots: control mice. **a** $p = 0.0016$, $CI = (-28.45, -9.28)$; **b** $p = 0.0433$, $CI = (-120.08, -2.58)$; **c** $p = 0.0002$, $CI = (-980.15, -623.18)$; **d** $p = 0.0038$, $CI = (13.88, 38.79)$; **e** $p = 0.0298$, $CI = (-0.26, 6.21)$; **f** $p = 0.0363$, $CI = (2.72, 64.75)$; **g** $p = 0.1374$, $CI = (-20.48, 5.15)$; **h** $p = 0.0176$, $CI = (1341.47, 5039.20)$; **i** $p = 0.0095$, $CI = (-0.99, -0.22)$. $DI$ = Mean difference confidence interval.

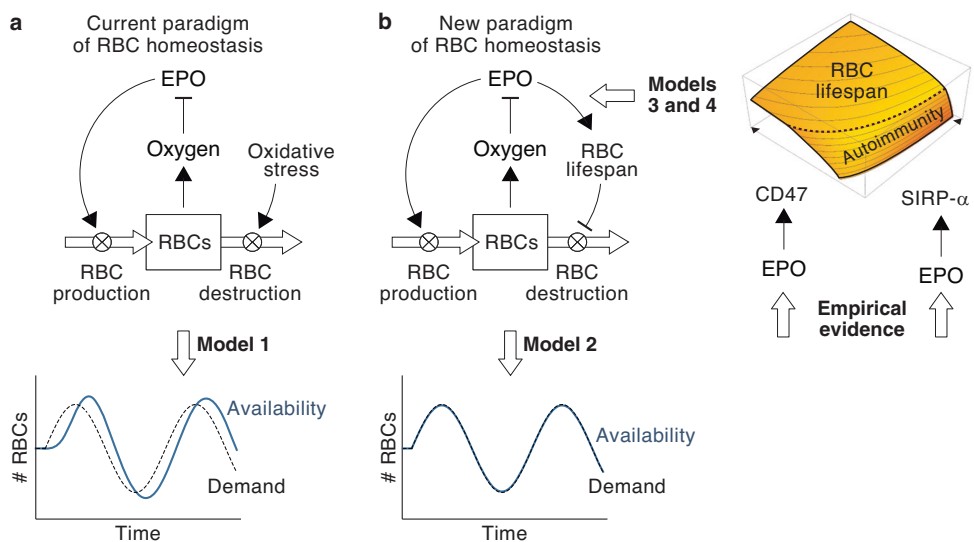

**Fig. 6 A model of RBC homeostasis. a** According to the current paradigm, EPO exclusively controls the rate of RBC production. Model 1 shows that this assumption leads to poor homeostatic control of RBC populations when the demand of the organism changes. **b** We suggest an alternative paradigm for RBC homeostasis in which EPO also controls the rate of RBC destruction by fine-tuning RBC lifespan. Model 2 shows that adaptive lifespans provide better homeostatic adaptations to changes in the needs of the body. Our data show that EPO controls RBC lifespan by regulating CD47 and SIRP-$\alpha$ expression in reticulocytes and macrophages, respectively.

$\alpha$ (e.g., using knockout mice) should lead to changes in the lifespan of circulating RBCs that could be compared to the predictions of our model. It is important to note that these are qualitative predictions based on a proposed mechanistic link between CD47 or SIRP-$\alpha$ expression, the production anti-RBC antibodies, and RBC lifespan. This mechanism transcends the particular choice of Equation (15) to simulate the evolution of CD47 and PS in the RBC membrane. Any other model producing qualitatively similar behaviors to that outlined in Fig. 6b would make the same predictions. In this regard, single-cell studies could prove useful to achieve a precise characterization of the dynamics of CD47 and PS in individual RBCs, as well as to identify potential sources of lifespan heterogeneity among cohorts of cells. This approach would provide a first step towards a more quantitative approach in the modeling of RBC lifespan regulation. At a systemic scale, forcing marked changes in the demand for RBCs in the organism (e.g., by inducing sharp fluctuations in oxygen availability) could allow to evaluate the inertia of the homeostatic mechanisms that control the number of RBCs in circulation, as well as the physiological range of RBC lifespan.

The view of RBC lifespan as an EPO-mediated homeostatic mechanism provides a sound explanation of neocytolysis, the destruction of young RBCs (aged only around 10 days) that occurs during de-acclimation from high altitudes[56–60]. Lower levels of partial pressures of atmospheric oxygen at high altitudes imply a greater demand for RBCs. Accordingly, EPO levels increase to raise the production of new cells[61]. Returning to sea level reverses the situation, creating a transient excess of RBCs in circulation[62]. It is widely accepted that neocytolysis is intended to rapidly reduce the number of cells in the blood in this situation[58,63–65].

The sharp reduction in RBC lifespan during neocytolysis is currently considered as a stress response triggered by high altitude hypoxia, which would make newly formed RBCs especially sensitive to oxidative damage[65–68]. For this reason, they would undergo accelerated senescence when oxygen levels return to normal[66]. From this viewpoint, hypoxia would create RBCs sensitive to oxidative stress in anticipation of an uncertain and unpredictable increase in oxygen levels. That a curtailed resistance to oxidation can be considered an active homeostatic mechanism is questionable. Creating RBCs prone to oxidative damage would be a puzzling response to hypoxia. Circulating RBCs are more necessary when oxygen availability is low. If anything, we should expect RBCs formed in hypoxia to exhibit greater resistance to failure and lower rates of functional senescence. As a matter of fact, it has been observed that RBCs formed under hypoxia exhibit longer lifespans[69].

From our approach, neocytolysis is not a stress response but the expected response to the changes in RBC demand during altitude acclimation and de-acclimation. At high altitudes, the increase in the concentrations of EPO would upregulate SIRP-$\alpha$ receptors in macrophages (Fig. 7a), lengthening lifespan to palliate the deficit of RBCs. The drop in EPO after returning to the sea would shorten RBC lifespan. This effect would be more pronounced in the newly formed RBCs that have very low levels of CD47 (Fig. 7b, c), so young RBCs could be phagocytized while older cells are spared (Fig. 7d). This is precisely what defines neocytolysis. In case that neocytolysis were not sufficient to achieve the needed contraction of the RBC population, the organism would resort to the production of anti-PS antibodies (Fig. 7c). The utility of these autoantibodies would be twofold. First, they would rapidly curtail the number of circulating RBCs. Second, the increase in the formation of free radicals under sustained hyperoxia would increase the likelihood of oxidative damage in the membrane of RBCs, so targeting PS in these circumstances would ensure the destruction of cells at greater risk of malfunction.

Our results reveal a rich regulatory landscape in which RBC homeostasis, an organism-level phenomenon, could rely on the interplay between molecules located on the membrane of RBCs and macrophages. This would give rise to a simple and powerful organism-level mechanism, capable of adjusting the number of RBCs in the wide range of configurations that can be adopted (both at evolutionary and ontogenetic scales) by mammalian bodies.

The modulation of RBC lifespan is subject to constrains both at cellular and systemic levels that delimit the functional range at which this homeostatic mechanism can operate. At a cellular scale, CD47 expression in newly formed RBCs cannot increase indefinitely, as occurs with the number of receptors present in individual macrophages, which sets an upper boundary to RBC lifespan. The flexibility of RBC lifespan also depends on systemic factors, such as the number of macrophages available to remove RBCs from the circulation. The phagocytic needs in case of a marked excess of RBCs might saturate the capacity of the system to maintain the required rate of RBC destruction, abnormally increasing the lifespan of circulating cells.

The capacity of the spleen and the liver to ensure an adequate rate of RBC destruction may also be compromised in pathological situations. This occurs, for instance, after massive transfusions, or in case of increased non-homeostatic destruction of circulating RBCs. Increased phagocytosis in these circumstances may cause the accumulation of iron in liver and splenic macrophages, leading to a form of cell death known as ferroptosis[70,71]. This, and other anomalous reductions in the number of macrophages may curtail the phagocytic capacity of the liver and spleen, interfering with normal lifespan determination and affecting the performance of the mechanisms of RBC homeostasis. This circumstance would create unwanted delays in the homeostatic response of the organism to changes in RBC demand. In this regard, it has been observed, both in mice and humans, that neutrophils efficiently remove opsonized RBCs in the spleen[72], suggesting that neutrophils complement the function of macrophages and autoimmunity to increase the phagocytic capacity of the organism when the number of RBCs in circulation must be rapidly reduced. The number of Kupffer cells has also been observed to increase in some circumstances, such as infections or alcohol-related diseases[73,74], implying that the number of macrophages available to remove RBCs could also be subject to EPO-mediated regulation to adjust the phagocytic rate to the homeostatic needs of the organism.

Despite its potential limitations, the adaptability of RBC lifespan would confer the organism with the ability to modulate the rate of RBC destruction in a wide range of normal circumstances, contributing to satisfy the variable demands for oxygen by the body tissues. A better understanding of how such flexibility is controlled under normal conditions can shed light on the origin of disorders associated to the number of RBCs. Thus, it has been observed that artificially reducing RBC lifespan, for instance using phenylhydrazine, leads to anemia[34]. Abnormal fluctuations of EPO leading to disruptions in RBC lifespan could also be in the origin of anemia in such seemingly disparate situations as spatial flights or malaria infections[33,75].

## Methods

**Numerical simulations**. The numerical simulations of Models 1 and 2 shown in Fig. 1 were performed using Julia software (version 1.6).

**Statistics and reproducibility**. Experiments in Fig. 5i where performed twice, in Fig. 5j, k once, and the rest of the experiments three or more times. The statistical analyses of the data (T-Tests and Mean difference confident intervals) were performed with

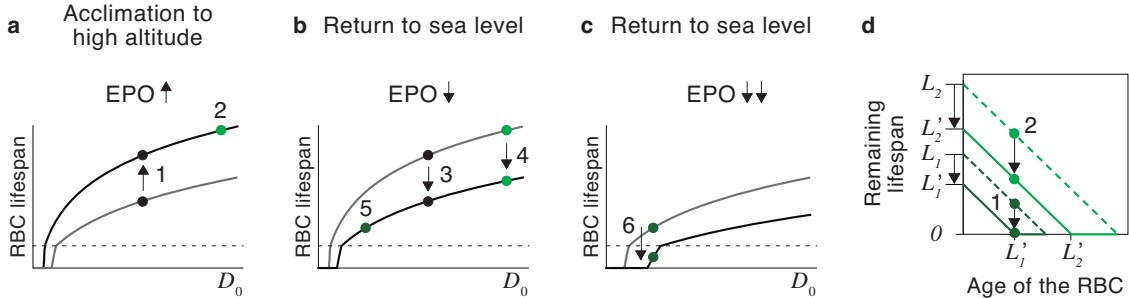

**Fig. 7 The origin of neocytolysis. a** The organism responds to lower oxygen availability at high altitudes by increasing the concentration of EPO. Within our paradigm, this would lengthen the lifespan of circulating RBCs (1), reducing their rate of destruction and contributing to increase the number of cells in the blood. New RBCs formed in these conditions would also exhibit longer predefined lifespans (2). **b** Returning to sea level would entail a drop in the concentration of EPO. According to our model, this would reduce RBC lifespan (3 and 4) and cause the formation of short-lived RBCs (5). Therefore, cohorts of RBCs that differ in their expected lifespan would coexist in the blood. **c** A sharp decrease in EPO levels could cause the destruction of RBCs through the immune phagocytosis, triggering an anti-RBC response, which would further reduce the lifespan of circulating cells. **d** The reduction in lifespan caused by the return to normal oxygen levels entails the instantaneous destruction of RBCs that have reached a critical age. This age threshold is not absolute but relative to each cohort of RBCs depending on their expected lifespan. Short-lived die when their expected lifespan falls from $L_1$ to $L_1'$ (1) whereas long-lived RBCs of the same age continue to circulate in the blood (2).

Wolfram Mathematica. Sample sizes ($n \geq 3$ in all cases) are indicated in the legend of Fig. 5.

**Mice and cells**. Animal experiments were performed according to EU regulations and approved by the Órgão Responsável pelo Bem-Estar Animal (ORBEA) of Instituto de Medicina Molecular and by the Direcção-Geral de Alimentação e Veterinária (Portugal). Female BALB/cByJ (age 6–8 weeks) were acquired from Charles River® Laboratories (Barcelona, Spain) and housed in groups of five in ventilated cages (IVCs) in the Rodent Facility of the Instituto de Medicina Molecular João Lobo Antunes-Lisboa. In each experiment, control mice and treated mice were of the same age and from the same batch. Rat Kupffer cells (Thermo-Fisher, RTKCCS) were cultured according to the manufacturer. In brief, the Kupffer cells were thawed and cultured in RPMI medium supplemented with 10% FBS, 1% Penicillin-Streptomycin, and 1% L-glutamine for 24 h in 96-wells plates pre-coated with collagen Type I (Sigma Aldrich).

**Double in vivo biotinylation staining of RBCs**. Six days after hypoxia treatment, control and treated mice were injected intravenously (i.v.) 2 consecutive days with 1 mg of biotin-X-NHS Ester (BXN) (Millipore, Merck) per mouse dissolved in 200 µl of saline buffer. Seven days later, mice were injected i.v. with 0.6 mg of BXN per mouse dissolved in 200 µl of saline buffer.

**EPO injections and staining of CD47**. Balb/c mice were injected subcutaneously (s.c.) with 40IU of EPO (Recombinant HuEpo-alpha, Gibco) for 4 consecutive days. We analyzed the expression of CD47 (phycoerythrin-conjugated anti-mouse CD47 antibody, Biolegend) in reticulocytes (CD71+, APC conjugated anti-mouse CD71, Biolegend) after the last EPO injection by flow cytometer (BD LSRFortessa). Blood was collected by a gentle puncture in the lateral tail vein of the mice using a 25 G needle. Blood drops were collected using a 2 µl automatic pipette and put into a collection tube with a heparin solution (1UI/ml in PBS 1X). Blood flow was stopped by applying finger pressure for approximately 30 s before returning the animal to its cage. All antibodies were used at 1 µg/µl for 30 min. at 4 °C unless specified otherwise. The FACS data were analyzed on FlowJo® software (TreeStar, v. 10.7.1). All experiments were performed at the Flow Cytometry Facility of the Insitituo de Medicina Molecular João Lobo Antunes.

**Monitoring of RBC lifespan after hyperoxia**. A hyperbaric chamber (ProOx 110 oxygen controller, BioSpherix, USA) was kindly provided by the laboratory of Dr. Claudio Franco at Instituto de Medicina Molecular (Lisboa). Mice were kept there for 5 days in a hyperoxic atmosphere (75% $O_2$). After that period, mice returned to normoxia conditions. At this moment, two initial biotin injections (1mg/200 µl) stained all the RBCs that were already in circulation before the hyperoxia condition (termed "biotin high RBCs" in the text). Seven days later, a third biotin injection (0.6 mg/200 µl) labeled all the newborn RBCs formed during the hyperoxia treatment (referred to as "biotin intermediate RBCs" in the text). Both populations were monitored once a week for 31 days by flow cytometer (BD LSRFortessa). To that end, 1 µl of blood was collected from the tail of each mouse by poking with a 25g needle, and stained with allophycocyanin (APC) conjugated streptavidin (Biolegend), which binds the biotin.

**Evaluation of SIRP-α expression in macrophages under hyperoxia**. BALB/cByJ mice were put in the hyperbaric chamber for 1.5 h in a hyperoxic atmosphere (75% $O_2$). Mice were sacrificed immediately through $CO_2$ narcosis, followed by the rapid exposure of the abdominal cavity and cannulation of the hepatic portal vein using a 26-gauge needle. After that, the inferior vena cava was immediately incisioned to enable drainage. 10mL of PBS 1X per mouse were perfused in total. Then, the livers were collected in Petri dishes with PBS 1X at 3% of FBS. The livers were cut into small pieces with scissors and put in 1mL of collagenase solution (0,5 mg of collagenase type IV/1 ml of RPMI 1640 medium with Type IV DNase I; final concentration: 25units/ml, Sigma-Aldrich), and incubated for 10 min at 37 °C. After stopping the reaction and passing the liver pieces through a strainer, the cells were washed and centrifuged at 50 g or 200 rpm for 3 min to discard the hepatocytes. The supernatant was centrifuged and osmotic lysis buffer of RBCs (ammonium chloride/potassium hydrogen carbonate buffer) was added over the pellet. After washing, the cells were stained with F4/80 (using FITC-conjugated anti-mouse antibody, Biolegend) and SIRP-α (PE-conjugated anti-mouse antibody, Biolegend). Both antibodies were used at 2 µg/µl.

**Evaluation of anti-phosphatidylserine antibodies**. BALB/cByJ mice were kept in the hyperbaric chamber for a period of 5 days in a hyperoxic atmosphere (75% $O_2$). After that, mice were

returned to normoxic conditions for 5 more days. Then, the mice were sacrificed by $CO_2$ narcosis, and blood was collected by cardiac puncture. Blood was clotted at room temperature (RT) for 30 min, followed by centrifugation at 12.000 rpm for 10 min at 4 °C. A commercial kit (Orgentec) was used to detect the presence of anti-PS antibodies in the serum of mice, following the instructions of the manufacturer. In brief, the samples were diluted 1/200 in the dilution buffer and 100 µl were added to each well and incubated for 1 h at RT. The wells were then washed 5 times with 300 µl of washing buffer, and then the samples were incubated with 100 ul of conjugate (goat anti-mouse HRP, Thermofisher) for 15 min at RT. Optical density was determined using a microplate reader (TECAN Infinite M200) at 450nM.

**Kupffer cells staining**. Kupffer cells cultured in 96-well plates were supplemented with 30 µg EPO per well (recombinant rat EPO carrier-free, Biolegend). After 1.5 h, they were collected after a 3 min incubation with trypsin and washed with cold PBS 1X. Then, they were stained with phycoerythrin-conjugated SIRP-α antibody (anti-rat, Biolegend) at 2 µg/µl. The Kupffer cells were kept on ice during the whole process.

**Phagocytosis assay in vitro**. Mouse splenocyte suspensions were obtained by mechanical disruption of the spleens through a cell strainer followed by osmotic lysis of RBCs with ammonium chloride/potassium hydrogen carbonate buffer. RBCs were labeled with 1 µg/µl DDAO (Invitrogen) for 1 h at 37 °C. Splenocytes were incubated for 1 h at 37 °C in 96-well plates in RPMI 1640 medium (3 ml) (Sigma-Aldrich) with or without EPO (35U/well) (recombinant mouse EPO carrier-free, Biolegend). The ratio of splenocytes to RBCs was 1:1. This ratio ensures non-saturating conditions for phagocytosis by splenocytes, as can be monitored by the high fluorescence peaks. Cells were transferred to ice before staining for macrophage and FAC analysis. To analyze macrophages in the phagocytosis assay, splenocytes were labeled with FITC-anti-F4/80 at 2 µg/µl.

**Reporting summary**. Further information on research design is available in the Nature Portfolio Reporting Summary linked to this article.

## Data availability
The data used in Fig. 5 are available in Supplementary Data 1. All data are available from the corresponding authors.

## Code availability
The code used in the numerical simulations of Model 2 is available in Supplementary Data 2. Supplementary Software contains two Jupyter notebooks corresponding to the code used for the simulation of changes in cell demand and hemorrhages, respectively (DOI 10.5281/zenodo.10370925). A PDF version of the code is also provided in Supplementary Data 2 (Supplementary Data 2.A and B include the models for changes in cell demand and hemorrhages, respectively).

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

## Acknowledgements
F.B. and C.F.A. are grateful to the Roechling Foundation for its support. Cr.F.A. and N.V.-L. were partially supported by the FCT grant no. EXPL/BIA-BIO-0644/2021. Cr.F.A. was partially supported by the MINECO grant PID2022-138187OB-I00. We thank the Flow Cytometry Facility of the Instituto de Medicina Molecular João Lobo Antunes and Daniela Ramalho for their technical support.

## Author contributions
Conceptualization: C.F.A. and Cr.F.A.; Formal analysis: C.F.A., Cr.F.A, F.B., and F.J.A.; Funding acquisition: Cr.F.A.; Investigation: C.F.A., N.L.V., F.B., S.M., F.J.A., and Cr.F.A.; Experimental work: Cr.F.A., N.L.V., F.B., S.M.; Modeling: C.F.A.; Writing (original draft) C.F.A. and Cr.F.A.; Writing (Review and editing): C.F.A., N.L.V., F.B., S.M., F.J.A., and Cr.F.A.

## Competing interests
The authors declare no competing interests.
