## [Peer review file · Communications Biology]

Reviewers' comments:

Reviewer #1 (Remarks to the Author):

In this manuscript, the authors propose a new role for erythropoietin (EPO) in the regulation of red blood cell (RBC) count. It is well established that RBC production is regulated by EPO, through a negative feedback loop mediated by oxygen levels. Maintaining and re-establishing RBC homeostasis during disease or injuries is of crucial importance for survival. Because of their relatively long lifespans, RBC production is too slow compensating changes in needs. For a 100 day lifespan, the RBC turnover rate is only 1% per day, too slow for compensation due to e.g. bleeding. The authors propose that EPO also act on RBC lifespan, by modulating macrophage-RBC interaction. Under this proposal, lifespan could be lengthened during times of need, and shortened during surplus. The authors propose two mechanisms based on CD47 and SIRP- α expression that modulates RBC 'programmed' lifespan, and by autoimmune responses that target circulating RBCs.

The manuscript presents interesting and convincing arguments for the existence of RBC lifespan regulation by EPO. However, despite the introduction of computational and mathematical models to support this notion, I feel that the manuscript lacks quantitative modelling evidence for the role of RBC lifespan in maintaining homeostasis. A cell population version of the models 3 and 4 would be useful to test the proposed mechanisms of EPO action on RBC lifespan and homeostasis. In particular, stability (ensuring that RBC count returns quickly and remains close to target) and robustness (with respect to model parameters) might be a challenge to achieve.

Theoretical perspective

1) In case of RBC surplus, production is no an issue, and extra RBCs can be destroyed, in an age-dependent manner, or without distinction for cell age. However, in case of need, lifespan cannot increase faster than time. Therefore, after a 10% loss in RBC, and assuming a 1% turnover per day, it would take 10 days to recover the total RBC count. Increasing RBC production can in principle be much faster.

2) The alternative model for effective RBC homeostasis is by regulation of RBC precursors. EPO favors proliferation and survival of erythroid progenitors, so in theory, release of new RBC in circulation can increase much faster than what is proposed here.

Comments on the model implementation

3) I tried to reproduce Figure 1, but couldn't match several panels, especially panels i) to l). In panels i) and j), RBC count is starting at 90% of the target, and increases back to target within 100 days (I assume time is in days). This does not seem to be possible given the change in production shown in panel j). At best, production is increased by from 1.0 to 1.1, i.e. an 10% increase. It would then require more than 100 days to recover the initial 10% loss. My own simulations show that the RBC count is back at target after more than 100 days.

4) Likewise, I cannot reproduce the sharp increase in RBC as shown in panel k), yellow curve. Even with a maximal increase in lifespan, the RBC count cannot increase that fast.

5) Moreover, the model as presented does not offer a stable production rate $p(t)$, when $\mu=0$. After hemorrhage, production goes up, but there is not guarantee it will go back to baseline: $p(t)$ will be constant whenever r and K are equal for long enough. This is clearly a problem that should be addressed.

6) The model for the lifespan L is difficult to justify. As presented, the lifespan $L(t)$ obeys a second-order equation that can be understood as a forced spring. This make L very prone to high-frequency oscillations when the friction coefficient σ is small. Figure 1 does not show such oscillations, suggesting that I missed some details in the model implementation.

7) I can reproduce qualitatively panels e) to h), but not perfectly.

8) Models 3 and 4 are descriptive at best. It would have been useful to validate the concept with a population model incorporating models 3 and 4. This is important, because these models introduce heterogeneity in cellular lifespan. Given that lifespan regulation is the main focus of the paper, the authors need to show that their regulation mechanisms is robust again heterogeneity in the lifespan of the different RBC cohorts (in addition to other sources of heterogeneity). Cohort with markedly different lifespans will induce large fluctuations in RBC counts that will likely promote further lifespan heterogeneities.

9) In Fig 4a, it looks like increasing and decreasing EPO concentrations have symmetrical effects, but this cannot be correct if $L' < 1$. Figure 4 therefore proposes a model that might run in contradiction with the constraint that $L' < 1$.

Reviewer #2 (Remarks to the Author):

The paper by C.F. Arias et al proposes a new mechanisms of homeostatic and adaptative regulation of erythropoiesis in mammals by which the lifespan of mature erythrocytes gets regulated by Epo. They first makes the point, using mathematical models for erythropoiesis that a regulatory mechanism that would only control erythrocytes production would not be optimal. They then propose mechanisms of Epo-driven erythrocytes lifespan control and finally demonstrates that some of their hypothesis can be verified experimentally in mice.

Although the message is interesting, and the effort of combining mathematical modeling with biological experiments is worthy of being highlighted, the manuscript in its current form suffers from a number of issues that need to be addressed before considering publication.

Major

1. My main concern is the fact that the authors seem to ignore a large part of the literature that leads them to claim novelty where in fact there are prior work that should be cited and taken into account. The most disturbing is their claim that they demonstrate that a fixed lifespan may pose a problem (Figure 1). Quoting from the following reference (doi:10.1016/j.jtbi.2007.09.041): "One can see that the simulation of model (4) gives better results when changes in erythrocytes mortality are taken into account." At the time the authors proposed that phenylhydrazine might be responsible for altering this erythrocyte lifespan, but the proposed Epo-driven mechanism would also perfectly do the job. In any case this should be taken into account.
2. Similarly, the idea that Epo might alter erythrocyte lifespan is not new and has been proposed earlier (10.1097/01.ASN.0000093253.42641.C1). This should also be cited and discussed.
3. In regards to points 1 and 2, the author should restrain from sensationalized claims such as: "(our work) implies a radical shift away from the current paradigm ". This is clearly an overstatement. Their view that Epo-driven lifespan regulation might play a role is interesting, but

as the authors claim themselves "the full validation of this model of homeostatic regulation of RBC lifespan will require further empirical studies". There are a couple of statement that calls for caution like "whether flexible and adaptive RBC lifespans are within reach of the organism". The author convincingly demonstrate that a flexible and adaptive RBC lifespans are within reach of the model, but this is far from a demonstration that this is indeed the case for the organism.

4. Both the code and the data should be made available at the beginning of the review process. As stated on the journal website: "All published manuscripts reporting original research in Nature Portfolio journals must include a data availability statement."

5. To the best of my understanding, equation 4 describes a linear production rate for erythrocytes. This is fine since the focus is made on the degradation of erythrocytes, but once again this modeling choice should be justified and the literature that proposes more sophisticated models of how Epo regulates erythrocytes production should be cited.

Minor:

1. Some equations are not numbered

2. Some lines are not numbered

3. Equation 6 : only an increase or decrease is modeled. This is a strong modeling choice to not model the fact that molecular variations will actually results from a net balance between a production and a degradation rate. This choice should be justified

4. Is Figure 2 really a simulation? If so both axes should be numerically labelled properly and parameter values should be indicated.

5. Where do the parameter values used for Figure 1 come from? What s the time unit?

6. It is strange that in the introduction, the lifespan of erythrocytes from many different species is given but not the murine one that will be used in the experiments.

7. The authors use reticulocytes (line 470). How were those obtained?

8. The statement that "mammalian RBCs are devoid of most intracellular structures [5, 6], so they escape the canonical way to remove cells from tissues, which is apoptosis [7, 8]." is only partly true. This is untrue for mammalian fetal erythrocytes, and is also untrue for many non-mammalian species that are not devoided of a nucleus. Furthermore an apoptosis-like cell death has been described in the references [7, 8] cited by the authors. This statement should therefore be rephrased (especially in light of major comment 2 upper).

9. The choice of a deterministic model precludes any study in the cell-to-cell variability. This should be made clear and discussed as a limitation, especially in light of the wealth of contemporary single-cell studies which have documented the magnitude and functional importance of said cell-to-cell variability.

10. The "eat/don't-eat" wording is not scientific and should be replaced accordingly.

11. Has the normality and heteroscedasticity of the data been verified before applying a T-test?

12. The number of repeats should be indicated in the legend to the Figure 5.

We thank the Reviewers for their constructive comments. Please find below the responses (in blue) to the Reviewer's comments (in black).

Reviewer #1 (Remarks to the Author):

In this manuscript, the authors propose a new role for erythropoietin (EPO) in the regulation of red blood cell (RBC) count. It is well established that RBC production is regulated by EPO, through a negative feedback loop mediated by oxygen levels. Maintaining and re-establishing RBC homeostasis during disease or injuries is of crucial importance for survival. Because of their relatively long lifespans, RBC production is too slow compensating changes in needs. For a 100 day lifespan, the RBC turnover rate is only 1% per day, too slow for compensation due to e.g. bleeding. The author propose that EPO also act on RBC lifespan, by modulating macrophage-RBC interaction. Under this proposal, lifespan could be lengthened during times of need, and shortened during surplus. The authors propose two mechanisms based on CD47 and SIRP-alpha expression that modulates RBC 'programmed' lifespan, and by autoimmune responses that target circulating RBCs.

The manuscript presents interesting and convincing arguments for the existence of RBC lifespan regulation by EPO. However, despite the introduction of computational and mathematical models to support this notion, I feel that the manuscript lacks quantitative modelling evidence for the role of RBC lifespan in maintaining homeostasis. A cell population version of the models 3 and 4 would be useful to test the proposed mechanisms of EPO action on RBC lifespan and homeostasis. In particular, stability (ensuring that RBC count returns quickly and remains close to target) and robustness (with respect to model parameters) might be a challenge to achieve.

We thank the Reviewer for his/her insightful comment. In fact, such a model would be very interesting indeed. We thought about that as well, but the problem is that this model would involve an agent-based approach, which is an altogether different modeling strategy than the one suggested in this work. Models 3 and 4 however, already show that lifespan may change depending on the dynamics of membrane signals. The consequences of a variable lifespan on homeostasis are illustrated with a population dynamics model (Model 1).

Theoretical perspective

- 1) In case of RBC surplus, production is no an issue, and extra RBCs can be destroyed, in an age-dependent manner, or without distinction for cell age. However, in case of need, lifespan cannot increase faster than time.

We thank the Reviewer for this remark, since it has made us realize that lifespan can actually increase faster than time. To clarify this point, it can be illustrative to make a thought experiment. Let us imagine, for instance, that cells become immortal at a given time t . In this case, lifespan becomes instantly infinite. This does not imply a sudden burst in the population, only that the "natural" rate of age-dependent cell death becomes zero. Cells could still die from other mortality causes, and the dynamics of the population in this case would depend entirely on the rate of cell production and this rate of external mortality. (Incidentally, this is the case of tumors.)

We have changed equation 12 and Model 2 to remove the constrain on positive increases in lifespan.

- 2) Therefore, after a 10% loss in RBC, and assuming a 1% turnover per day, it would take 10 days to recover the total RBC count. Increasing RBC production can in principle be much faster.

Our model has been deduced from first principles, and theoretical concerns should refer to this deduction. Although we understand that we almost cannot avoid to make this type of calculations, extra care should be adopted when dealing with such complex dynamics.

In addition, the Reviewer is using a different model in which cell turnover is constant after a 10% cell loss. In our model, production depends on the difference between the current number of cells and RBC demand, as indicated in equation 10. Therefore, there is no space for a fixed 1% turnover per day.

We would like also to point out that increased lifespan and increased production are not mutually exclusive in our model, quite the opposite. The logic of cell production is identical in the models with constant and variable lifespans. Regardless of how fast production increases, if lifespan increases too, then the rate of cell death decreases, which accelerates the recovery of the population.

2) The alternative model for effective RBC homeostasis is by regulation of RBC precursors. EPO favors proliferation and survival of erythroid progenitors, so in theory, release of new RBC in circulation can increase much faster than what is proposed here.

We thank the Reviewer for this comment, which allows us to clarify this specific point. Our model of RBC production does not refer to any particular mechanism of RBC production. It uses an abstract logic to link cell production with cell deficit (see Equation 10). In this regard, the proliferation of erythroid precursors and any other mechanism to increase the number of cells is implicitly included in the model. We have made this point clearer in the text:

"The regulation of RBC production is a complex process in which EPO levels control the proliferation and differentiation of erythroid precursors in the bone marrow. From a modeling perspective, the role of EPO can be implemented either explicitly (see for instance [38, 39]), or implicitly, through its resultant effects on the formation of new RBCs (as in [40]). For the sake of simplicity, we will use the latter strategy, and will not explicitly consider the dynamics of EPO or its effect on precursor cells. Despite its mechanistic complexity, the underlying logic of EPO regulation can be described in simple terms: it increases production when the number of cells is below the needs of the body and decreases production otherwise."

It is possible to use a different equation for RBC production in the model. It suffices to change Equation 10 for a different one. However, it is important to insist that increased lifespan does not exclude increased production. Whatever the equation of RBC production, it is the same for constant and variable lifespan. A faster production would increase the rate of cell formation in a population with constant lifespan and also in a population with variable lifespan. In the latter, if lifespan increases, the rate of cell mortality decreases, which contributes to increasing the number of cells regardless of production.

Comments on the model implementation

3) I tried to reproduce Figure 1, but couldn't match several panels, especially panels i) to l). In panels i) and j), RBC count is starting at 90% of the target, and increases back to target within 100 days (I assume time is in days). This does not seem to be possible given the change in production shown in panel j). At best, production is increased by from 1.0 to 1.1, i.e. an 10% increase. It would then require more than 100 days to recover the initial 10% loss. My own simulations show that the RBC count is back at target after more than 100 days.

The Reviewer has not shared his/her code, so it is difficult to guess where the problem with his/her implementation can be (more on this in the response to point 5). In any case, the Reviewer seems to be assuming that 10% increase in cell number and 10% increase in cell production are comparable. However, they are not. It is possible to have 100 cells at the steady state with a production of 10 cells/day or with a production of 100 cells/day, depending on the rate of cell mortality and on lifespan. A 10% increase entails 1 more cell/day in the first case and 10 in the second one. If the population drops from 100 to 90 cells, the recovery will be clearly different in both scenarios.

4) Likewise, I cannot reproduce the sharp increase in RBC as shown in panel k), yellow curve. Even with a maximal increase in lifespan, the RBC count cannot increase that fast.
See point 5.

5) Moreover, the model as presented does not offer a stable production rate $p(t)$, when $\mu=0$. After hemorrhage, production goes up, but there is not guarantee it will go back to baseline: $p(t)$ will be constant whenever r and K are equal for long enough. This is clearly a problem that should be addressed.

We apologize for not having included the code for numerical simulations of these models in our first submission.

The Reviewer has not shared his/her code, so it is difficult to guess where is the problem. However, we think that it could result from an error in the implementation of hemorrhages. We understand that the Reviewer has modeled hemorrhages by setting the initial population value at $0.9K$.

Hemorrhages cannot be simulated by just reducing the number of cells at the start of the simulation. It is also necessary to modify the history of the population. Otherwise, the cells that are lost in the hemorrhage "continue to die" of old age when they reach their lifespan. This adds an artificial source of cell mortality that interferes with the dynamics of the system.

When this issue is taken into account, both the number of cells and cell production (with constant and variable lifespans) return to baseline:

If this issue is not taken into account, production does not return to baseline, which coincides with the result obtained by the Reviewer:

We think that this may be the problem with the Reviewer's code. This error also explains the remark in the previous point. The Reviewer's model incurs in an additional cost of spurious mortality.

We have included as Supplementary Material a Jupyter notebook (as well as a PDF version) with the details of the model implementation.

6) The model for the lifespan L is difficult to justify. As presented, the lifespan $L(t)$ obeys a second-order equation that can be understood as a forced spring. This makes L very prone to high-frequency oscillations when the friction coefficient σ is small. Figure 1 does not show such oscillations, suggesting that I missed some details in the model implementation.

We do not agree with the Reviewer in this point. To model the scenario of variable lifespan, we assume that lifespan can oscillate around a reference value. An oscillator is a natural choice to model this behavior. Furthermore, friction naturally simulates the resistance of the system to change lifespan. This resistance has a great impact in the dynamics of the population. It is difficult to think of a better modeling framework for our hypothesis.

7) I can reproduce qualitatively panels e) to h), but not perfectly.

We hope that the new Supplementary Material will be useful.

8) Models 3 and 4 are descriptive at best. It would have been useful to validate the concept with a population model incorporating models 3 and 4.

In fact, these models intend to qualitatively describe the effect of membrane signals on RBC lifespan. The model suggested by the Reviewer would require an agent-based approach, which is beyond the scope of this work.

This is important, because these models introduce heterogeneity in cellular lifespan. Given that lifespan regulation is the main focus of the paper, the authors need to show that their regulation mechanisms are robust against heterogeneity in the lifespan of the different RBC cohorts (in addition to other sources of heterogeneity). Cohort with markedly different lifespans will induce large fluctuations in RBC counts that will likely promote further lifespan heterogeneities.

This interesting hypothesis is one of the multiple lines in which our model could be further developed in the future.

9) In Fig 4a, it looks like increasing and decreasing EPO concentrations have symmetrical effects, but this cannot be correct if $L' < 1$. Figure 4 therefore proposes a model that might run in contradiction with the constraint that $L' < 1$.

Again, we thank the Reviewer for making us realize that the constraint $L' < 1$ was a wrong assumption. We have removed it from the equations of our model.

Reviewer #2 (Remarks to the Author):

The paper by C.F. Arias et al proposes a new mechanisms of homeostatic and adaptative regulation of erythropoiesis in mammals by which the lifespan of mature erythrocytes gets regulated by Epo. They first makes the point, using mathematical models for erythropoiesis that a regulatory mechanism that would only control erythrocytes production would not be optimal. They then propose mechanisms of Epo-driven erythrocytes lifespan control and finally demonstrates that some of their hypothesis can be verified experimentally in mice.

Although the message is interesting, and the effort of combining mathematical modeling with biological experiments is worthy of being highlighted, the manuscript in its current form suffers from a number of issues that need to be addressed before considering publication.

Major

1. My main concern is the fact that the authors seem to ignore a large part of the literature that leads them to claim novelty where in fact there are prior work that should be cited and taken into account. The most disturbing is their claim that they demonstrate that a fixed lifespan may pose a problem (Figure 1). Quoting from the following reference (doi:10.1016/j.jtbi.2007.09.041): "One can see that the simulation of model (4) gives better results when changes in erythrocytes mortality are taken into account." At the time the authors proposed that phenylhydrazine might be responsible for altering this erythrocyte lifespan, but the proposed Epo-driven mechanism would also perfectly do the job. In any case this should be taken into account.

We thank the Reviewer for this reference, which we did not know. However, we disagree with the Reviewer's interpretation of these results and its relationship with our work.

This paper presents a model of erythropoiesis that "takes a positive feedback of erythrocytes on progenitor apoptosis into account, and incorporates a negative feedback of erythrocytes on progenitor self-renewal." In this model, RBC mortality **is considered a constant**: "Erythrocytes are assumed to die **with a constant rate gamma**." It is important to stress that in this model, RBC mortality does not change in the course of each simulation. The basic assumption of our work is just the opposite: RBC lifespan is not constant and changes dynamically depending on the needs of the organism.

The quote mentioned by the referee has to be put in context. In the article, we can read "Fig. 7. Evolution of hematocrit over 44 days (Panel A), with values given in Table 1, except $\gamma=0.15 \text{ d}^{-1}$." (The value of parameter gamma in Table 1 is 0.025 d^{-1} .) The authors are simulating the consequences of anemia on hematocrit. This is why increasing RBC mortality yields better results.

In summary, in the phrase "One can see that the simulation of model (4) gives better results when changes in erythrocytes mortality are taken into account.", changes refer to alternative scenarios that differ in the value of RBC mortality (0.15 d^{-1} vs. 0.025 d^{-1}). This rate remains constant in both scenarios. We insist, the authors are not comparing a constant lifespan with a variable one, but two cases in which lifespan is constant albeit different.

2. Similarly, the idea that Epo might alter erythrocyte lifespan is not new and has been proposed earlier (10.1097/01.ASN.0000093253.42641.C1). This should also be cited and discussed.

We thank the Reviewer for calling our attention on this article. We have included this reference (#37) in Section "Effect of constant and adaptive lifespan on the homeostasis of red blood cells":

"The question arises of whether adaptive RBC lifespans are within reach of the organism. Published evidence suggests that this might be the case. In particular, it has been observed that EPO may bind

circulating RBCs, thereby improving their viability, which would increase their lifespan and reduce the rate of cell destruction in the blood [37]. In the following sections we will explore other mechanisms that could participate in the regulation of RBC lifespan by EPO."

And in the Discussion:

"Previous studies have shown that EPO might increase RBC lifespan by delaying senescence [37]. We suggest that EPO could also regulate RBC lifespan through its effects on CD47 and SIRP- α expression and through EPO-induced autoimmunity."

3. In regards to points 1 and 2, the author should restrain from sensationalized claims such as: "(our work) implies a radical shift away from the current paradigm ". This is clearly an overstatement. Their view that Epo-driven lifespan regulation might play a role is interesting, but as the authors claim themselves "the full validation of this model of homeostatic regulation of RBC lifespan will require further empirical studies".

We have rephrased these statements throughout the text to soften the tone.

There are a couple of statement that calls for caution like "whether flexible and adaptive RBC lifespans are within reach of the organism". The author convincingly demonstrate that a flexible and adaptive RBC lifespans are within reach of the model, but this is far from a demonstration that this is indeed the case for the organism.

In fact, we have not proven that a flexible lifespan is within reach of the model. It is a model assumption, not a deduction. In addition, the phrase is not used to suggest that we have proven that the organism uses adaptive lifespans. It is only used in Section "Effect of constant and adaptive lifespan on the homeostasis of red blood cells" to raise a question that we deem interesting:

"The question arises of whether adaptive RBC lifespans are within reach of the organism."

4. Both the code and the data should be made available at the beginning of the review process. As stated on the journal website: "All published manuscripts reporting original research in Nature Portfolio journals must include a data availability statement."

We have included the code and the data used in Fig. 5 as supplementary material. We apologize for not having made them available in the first submission.

5. To the best of my understanding, equation 4 describes a linear production rate for erythrocytes. This is fine since the focus is made on the degradation of erythrocytes, but once again this modeling choice should be justified and the literature that proposes more sophisticated models of how Epo regulates erythrocytes production should be cited.

This is an interesting issue that was not sufficiently explained in the text, and we thank the Reviewer for this remark. We made this point more clear, including references to other published models:

"The regulation of RBC production is a complex process in which EPO levels control the proliferation and differentiation of erythroid precursors in the bone marrow. From a modeling perspective, the role of EPO can be implemented either explicitly (see for instance [38, 39]), or implicitly, through its resultant effects on the formation of new RBCs (as in [40]). For the sake of simplicity, we will use the latter strategy, and will not explicitly consider the dynamics of EPO or its effect on precursor cells. Despite its mechanistic complexity, the underlying logic of EPO regulation

can be described in simple terms: it increases production when the number of cells is below the needs of the body and decreases production otherwise."

Minor:

1. Some equations are not numbered

Corrected.

2. Some lines are not numbered

Corrected.

3. Equation 6 : only an increase or decrease is modeled. This is a strong modeling choice to not model the fact that molecular variations will actually results from a net balance between a production and a degradation rate. This choice should be justified

This issue is addressed in reference 33. We have made this point clearer in the text:

"(see [33] for further details about the rationale for the choice of these equations and the meaning of the parameters)"

According to the literature though, CD47 expression is greater in young RBCs than in old ones, whereas that of PS exhibits the opposite pattern. This implies that the net variation in CD47 is negative, and that of PS is positive. (More on this in point 9.)

4. Is Figure 2 really a simulation? If so both axes should be numerically labelled properly and parameter values should be indicated.

It is not a simulation actually. This figure represents the behavior of the functions shown in Equation 15, which are a straight line with negative slope and an exponential decay. We have made this point clear in the figure caption. This figure is intended to show qualitative behaviors and the parameter values have been chosen arbitrarily to that end. We believe that including these values, or numerically labelling the axes would only add complexity to the figure without providing any valuable additional information.

5. Where do the parameter values used for Figure 1 come from? What s the time unit?

Parameters have been chosen arbitrarily to illustrate the behavior of Models 1 and 2. The units are also arbitrary, and, in our opinion, they are not relevant to convey the intended information.

6. It is strange that in the introduction, the lifespan of erythrocytes from many different species is given but not the murine one that will be used in the experiments.

We have included this information based on a new reference (<https://doi.org/10.1182/blood.V13.8.789.789>). We do not understand why this is detail is qualified as strange by the Reviewer. We have just chosen a random set of species with the only intention to stress the fact that RBC lifespan differs among mammals.

7. The authors use reticulocytes (line 470). How were those obtained?

We do not understand this question. They were identified using CD71 as indicated in Methods.

8. The statement that “mammalian RBCs are devoid of most intracellular structures [5, 6], so they escape the canonical way to remove cells from tissues, which is apoptosis [7, 8].” is only partly true. This is untrue for mammalian fetal erythrocytes, and is also untrue for many non-mammalian species that are not devoided of a nucleus. Furthermore an apoptosis-like cell death has been described in the references [7, 8] cited by the authors. This statement should therefore be rephrased (especially in light of major comment 2 upper).

We thank the Reviewer for this remark. It led us to remove the whole phrase from the text. In fact, our statement gave the false impression that apoptotic-like RBC death was an alternative pathway to remove RBCs from the circulation when it is not. Regardless of the molecular details of senescence, RBCs are removed from the circulation by sinusoidal macrophages, and this is the relevant part of the process in our model.

9. The choice of a deterministic model precludes any study in the cell-to-cell variability. This should be made clear and discussed as a limitation, especially in light of the wealth of contemporary single-cell studies which have documented the magnitude and functional importance of said cell-to-cell variability.

We do not agree with the Reviewer in this point. The choice of a model does not preclude any future study using complementary models. Moreover, the proposed mechanistic link between molecules present in the membranes of RBCs and macrophages does not depend on the particular equations chosen in this work. We have included a few lines to make this point clear in the text:

"At a molecular scale, the experimental manipulation of key factors such as CD47 or SIRP- α (e.g. using knockout mice) should lead to changes in the lifespan of circulating RBCs that could be compared to the predictions of our model. It is important to note that these are qualitative predictions based on a proposed mechanistic link between CD47 or SIRP- α expression, the production anti-RBC antibodies, and RBC lifespan. This mechanism transcends the particular choice of Equations 15 to simulate the evolution of CD47 and PS in the RBC membrane. Any other model producing qualitatively similar behaviors to that outlined in Fig. 6.B would make the same predictions. In this regard, single-cell studies could prove useful to achieve a precise characterization of the dynamics of CD47 and PS in individual RBCs, as well as to identify potential sources of lifespan heterogeneity among cohorts of cells. This approach would provide a first step towards a more quantitative."

10. The “eat/don’t-eat” wording is not scientific and should be replaced accordingly.

These terms have long been used in the literature. Searching the expression "eat-me signals" in Google Scholar yields 23.000 results:

https://scholar.google.es/scholar?hl=es&as_sdt=0%2C5&q=%22eat-me%22+signals&btnG=

11. Has the normality and heteroscedasticity of the data been verified before applying a T-test?

We have not checked normality or heteroscedasticity. We acknowledge that our statistical analysis is weak but we were under severe restrictions regarding the use of mice.

12. The number of repeats should be indicated in the legend to the Figure 5.

We have included this information in the caption.

Reviewers' comments:

Reviewer #2 (Remarks to the Author):

While we appreciate the effort made for answering my concern, there are still a few issues that need to be addressed.

My main concern is the fact that the authors seem to ignore a large part of the literature that leads them to claim novelty where in fact there are prior work that should be cited and taken into account.

"we disagree with the Reviewer's interpretation of these results and its relationship with our work. (...) We insist, the authors are not comparing a constant lifespan with a variable one, but two cases in which lifespan is constant albeit different."

Yes I am aware of that. Nevertheless, the very possibility that a constant lifespan was not ideal dates back from the cited reference. I therefore still think it would be relevant to cite as the first occurrence that modeling erythropoiesis with a fixed lifespan might not be ideal.

5. Where do the parameter values used for Figure 1 come from? What s the time unit?

"Parameters have been chosen arbitrarily to illustrate the behavior of Models 1 and 2. The units are also arbitrary, and, in our opinion, they are not relevant to convey the intended information." This should be clearly stated in the Figure legend.

4. Is Figure 2 really a simulation? If so both axes should be numerically labelled properly and parameter values should be indicated.

"This figure is intended to show qualitative behaviors and the parameter values have been chosen arbitrarily to that end."

This should be clearly stated in the Figure legend.

7. The authors use reticulocytes (line 470). How were those obtained?

"We do not understand this question. They were identified using CD71 as indicated in Methods."

The question was: how were those reticulocytes obtained from mice? I would guess some blood was collected, but how? From what organ? Were the mice euthanized for that purpose? I guess this must have been described in order to be able to state that "Animal experiments were performed according to EU regulations"

11. Has the normality and heteroscedasticity of the data been verified before applying a T-test?

We have not checked normality or heteroscedasticity. We acknowledge that our statistical analysis is weak but we were under sever restrictions regarding the use of mice.

I do not see why the "severe restrictions regarding the use of mice" should prevent a correct statistical analysis. Eventhough the number of mice is small, normality and heteroscedasticity of the data must be verified before applying a T-test. If those conditions are not met, then a non parametric test should be used instead.

Please find below the responses (in blue) to the Reviewer's comments (in black).

Reviewer #2 (Remarks to the Author):

While we appreciate the effort made for answering my concern, there are still a few issues that need to be addressed.

My main concern is the fact that the authors seem to ignore a large part of the literature that leads them to claim novelty where in fact there are prior work that should be cited and taken into account. "we disagree with the Reviewer's interpretation of these results and its relationship with our work. (...) We insist, the authors are not comparing a constant lifespan with a variable one, but two cases in which lifespan is constant albeit different."

Yes I am aware of that. Nevertheless, the very possibility that a constant lifespan was not ideal dates back from the cited reference. I therefore still think it would be relevant to cite as the first occurrence that modeling erythropoiesis with a fixed lifespan might not be ideal.

We have included the reference suggested by the Reviewer. We have added this phrase in the Introduction:

"Previous studies have shown the effects that artificially changing lifespan has on the dynamics of RBC populations [34]. Our results suggest that the organism could adopt a similar strategy, varying RBC lifespan, and consequently the rate of RBC destruction, to adjust the population of circulating cells depending the conditions of oxygen availability. " (lines 47-50)

and in the Discussion:

"Thus, it has been observed that artificially reducing RBC lifespan, for instance using phenylhydrazine, leads to anemia [40]. " (lines 345-346)

5. Where do the parameter values used for Figure 1 come from? What s the time unit?

"Parameters have been chosen arbitrarily to illustrate the behavior of Models 1 and 2. The units are also arbitrary, and, in our opinion, they are not relevant to convey the intended information." This should be clearly stated in the Figure legend.

We thank the Reviewer for this suggestion, and we apologize for not having done it before.

4. Is Figure 2 really a simulation? If so both axes should be numerically labelled properly and parameter values should be indicated.

"This figure is intended to show qualitative behaviors and the parameter values have been chosen arbitrarily to that end."

This should be clearly stated in the Figure legend.

Done.

7. The authors use reticulocytes (line 470). How were those obtained?

"We do not understand this question. They were identified using CD71 as indicated in Methods."

The question was: how were those reticulocytes obtained from mice? I would guess some blood was collected, but how? From what organ? Were the mice euthanized for that purpose? I guess this must have been described in order to be able to state that "Animal experiments were performed according to EU regulations"

We apologize for the misunderstanding. We have included these lines in Methods:

"Blood was collected by a gentle puncture in the lateral tail vein of the mice using a 25 G needle. Blood drops were collected using a 2 μ l automatic pipette and put into a collection tube with a heparin solution (1UI/ml in PBS 1X). Blood flow was stopped by applying finger pressure for approximately 30 seconds before returning the animal to its cage." (line 546-549)

11. Has the normality and heteroscedasticity of the data been verified before applying a T-test?

We have not checked normality or heteroscedasticity. We acknowledge that our statistical analysis is weak but we were under severe restrictions regarding the use of mice.

I do not see why the "severe restrictions regarding the use of mice" should prevent a correct statistical analysis. Eventhough the number of mice is small, normality and heteroscedasticity of the data must be verified before applying a T-test. If those conditions are not met, then a non parametric test should be used instead.

The problem is that our sample sizes are too small for normality and heteroscedasticity tests to perform adequately. For instance, the Shapiro–Wilk test cannot even be computed in Wolfram Mathematica for such small samples.